# Exposure Completing for Temporally Consistent Neural High Dynamic Range Video Rendering

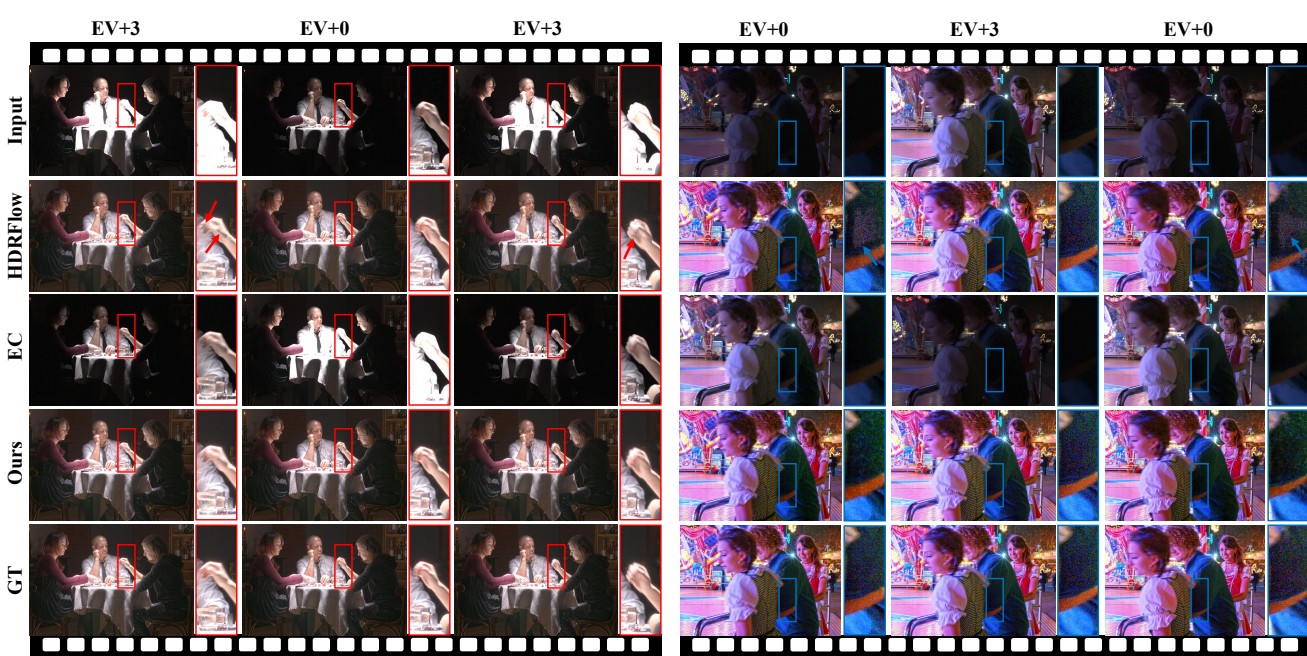

(a) Indoor scene with saturation and motions.      (b) Outdoor scene with noise and motions.

**Figure 1: Qualitative comparison between the proposed method and HDRFlow [40] on two different scenes. The input low dynamic range (LDR) videos consist of frames with alternate exposures, meaning that the exposure changes and part of exposure information is missing at every time stamp (1-st row). Prior arts, *e.g.*, the HDRFlow, struggle to achieve temporally consistent high dynamic range (HDR) results due to the exposure change and the motion that causes information loss (2-nd row). In contrast, the proposed method achieves temporally consistent HDR reconstruction results (4-th row) by completing the absent exposure frames (3-rd row) at each time stamp. "EV": the exposure value for each input LDR frame. "EC": our exposure completing results for the absent exposure corresponding to the input LDR frames.**

## ABSTRACT

High dynamic range (HDR) video rendering from low dynamic range (LDR) videos where frames are of alternate exposure encounters significant challenges, due to the exposure change and absence at each time stamp. The exposure change and absence make existing methods generate flickering HDR results. In this paper, we propose a novel paradigm to render HDR frames via completing the absent exposure information, hence the exposure information is complete and consistent. Our approach involves interpolating neighbor LDR frames in the time dimension to reconstruct LDR frames for the absent exposures. Combining the interpolated and given LDR frames, the complete set of exposure information is available at each time stamp. This benefits the fusing process for HDR results, reducing noise and ghosting artifacts therefore improving temporal consistency. Extensive experimental evaluations on standard benchmarks demonstrate that our method achieves state-of-the-art performance, highlighting the importance of absent exposure completing in HDR video rendering. The code will be made publicly available upon the acceptance of this paper.

## CCS CONCEPTS

• **Computing methodologies → Artificial intelligence**; **Computer vision**; **Image and video acquisition**; **Computational photography**.

*ACM MM, 2024, Melbourne, Australia*

© 2024 Copyright held by the owner/author(s). Publication rights licensed to ACM.
ACM ISBN 978-x-xxxx-xxxx-x/YY/MM
https://doi.org/10.1145/nnnnnnn.nnnnnnn

## KEYWORDS

HDR video, Temporal consistency, Exposure completing

## 1 INTRODUCTION

Compared with the high dynamic range (HDR) of natural scenes, the standard digital camera can only capture low dynamic range (LDR) information. To achieve the goal of "what you see is what you get" in photography, delicate optical systems [22, 33, 38] are designed to simultaneously capture LDR images with different exposures which cover the whole dynamic range to reconstruct HDR results. However, these systems are expensive and un-portable, making HDR contents not easily accessible for general users [18]. To address this limitation, computational methods [17, 18, 29] are designed to render HDR videos from LDR videos whose frames are exposed alternately with different exposures (as shown in 1-st row in Fig. 1).

In this setting, the incomplete exposure information varies along the time dimension. Existing neural HDR rendering methods [3, 6, 15, 40] align neighbor LDR frames according to the LDR frame of current time stamp (called reference frame in this paper) and fuse the aligned neighbor LDR frames into the reference frame to supplement the missing exposure information. The exposure of reference frame changes at every time stamp, which means that the reference frames of different exposures may have different defects: saturation for highly exposed LDR frames, and noise for lowly exposed ones. Due to that the methods mentioned above heavily depend on reference frames, HDR results from these methods may inherit defects of the reference frames, suffering from the artifacts, *i.e.*, the ghosting (2-nd row in Fig. 1(a)) and noise (2-nd row in Fig. 1(b)), thereby causing temporal inconsistency.

To tackle this problem, we revisit early traditional methods [17, 18] for HDR rendering and are motivated to complete the missing exposure information for each time stamp [17, 18]. This idea follows the philosophy of optical HDR systems [38] that are designed to obtain different exposed LDR frames at exact the same time. In this way, defects of different exposed frames can be mutually compensated. In this paper, we propose the Neural Exposure Completing HDR (NECHDR) framework to reconstruct the LDR frames with missing exposure information. In this way, full exposure information at each time stamp can be covered by the reference and completed LDR frames, therefore the exposure information is complete and consistent along the time dimension, which benefits the temporal consistency of the rendered HDR videos.

In the proposed NECHDR framework, pyramid features of input LDR frames are extracted by the feature encoder, then fed into the exposure completing decoder and the HDR rendering decoder. The exposure completing decoder interpolates the features of neighbor LDR frames at every level of the feature pyramid. The interpolated LDR features are combined with the features from input LDR images as the inputs to HDR rendering decoder. The HDR rendering decoder estimates coarse HDR results and the optical flows at every level. The flows can facilitate feature interpolating in exposure completing decoder. At the end of NECHDR, a simple blending network is used to integrate interpolated LDR frames, input frames, and coarse HDR frames, which can achieve high-quality and temporally consistent HDR reconstruction results.

Extensive experimental results on multiple public benchmarks demonstrate the superiority of the proposed NECHDR: by completing the missing exposure information (3-rd row in Fig. 1), our method mitigates ghosting resulting from large motions for the time stamps with highly exposed reference frames (4-th row in Fig. 1(a)), and reduces the noise level for the lowly exposed reference frames (4-th row in Fig. 1(b)), hence achieves better temporal consistency. Our work sheds light again on the exposure completing for HDR video rendering. The contributions can be summarized as follows:

• Our work firstly implements the idea of exposure completing for neural HDR rendering.

• Our work proposes a novel HDR video rendering framework, *a.k.a.*, the NECHDR, which completes the missing exposure information by interpolating LDR frames.

• Our NECHDR achieves new state-of-the-art performance on current benchmarks.

## 2 RELATED WORK

### 2.1 HDR Image Rendering

HDR imaging technology aims to extend the dynamic range of images by merging LDR images with different exposures. Both the HDR image and video tasks encounter the misalignment between LDR images. Pioneer studies for HDR image rendering employ image alignment techniques to address this issue. Early works [9, 12, 19, 24, 35, 47] often align LDR images globally, then detect and discard unaligned regions. Directly discarding these regions leads to significant information loss, posing challenges for HDR image rendering. To address these issues, optical-flow-based [1, 14, 49] and patch-based methods [13, 28, 37] are proposed. However, large motion still makes these methods fail to produce artifact-free HDR results. With the rapid development of deep learning, many studies for HDR image rendering switch their focus to neural networks [2, 5, 16, 25, 27, 34, 39, 42–44], achieving promising results. Kalantari *et al.* [16] propose using neural networks to align input LDR images with predicted optical flow. Wu *et al.* [39] directly map LDR images to HDR images, thereby avoiding alignment errors. Yan *et al.* [43, 45] use spatial attention to implicitly align LDR images, achieving further improvement. Liu [27] *et al.*introduce a ghost-free imaging model based on swin-transformer [26]. Yan *et al.* [42] utilize patch-based and pixel-based fusion to search for information to complement the reference frame from the other frames. However, these HDR image rendering methods assume a fixed exposure for reference frames, typically medium exposure, therefore is not suitable for HDR video rendering where reference frames have alternating exposures.

### 2.2 HDR Video Rendering

HDR video can be obtained by delicate optical systems, such as scan-line exposure/ISO [4, 11], internal/external beam splitter [23, 31, 38], modulo camera [48] and neuromorphic camera [10]. However, these expensive systems are hardly accessible to general users. Therefore, rendering HDR videos from LDR videos is investigated. Kang *et al.* [18] utilize both global and local alignment for reference frames with neighbor frames. Kalantari *et al.* [17] propose a patch-based optimization algorithm to reconstruct HDR video by synthesizing missing exposures for each frame. These traditional methods offer

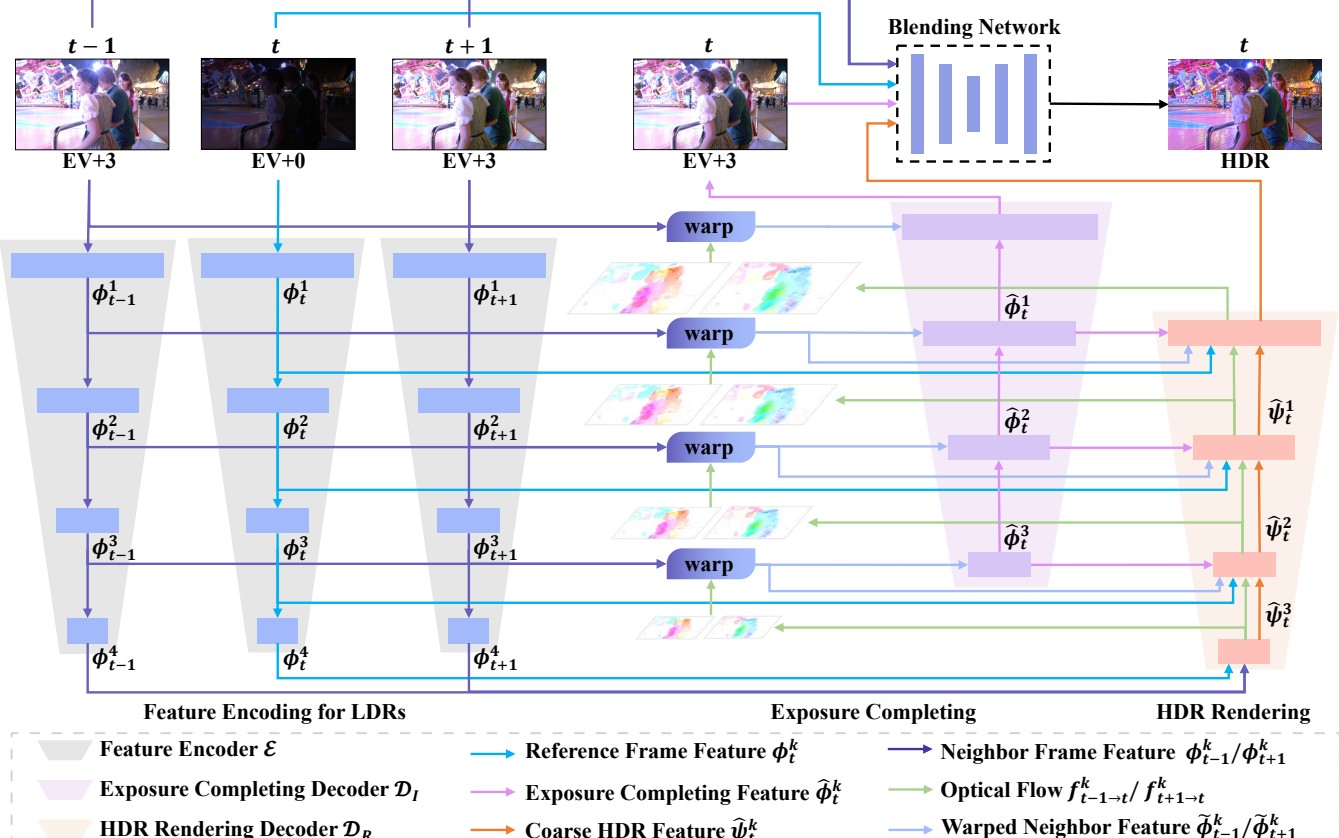

**Figure 2: Pipline of the proposed NECHDR network. Our network mainly consists of three processes: feature encoding for LDRs, exposure completing for LDR frames with missing exposures, and HDR rendering. We extract pyramid features from the input LDR frames using a parameter shared feature encoder. Then, the optical flows predicted by the lower-level HDR rendering decoder is used to warp neighbor frames to the reference frame. Subsequently, the exposure completing decoder performs exposure completing based on the warped neighbor frame features. The input features and the completed feature are fed into the HDR rendering decoder to render the coarse HDR frame. Finally, the original frames, the completed frame, and the coarse HDR frame are fused together through a simple blending network to produce a high-quality HDR result.**

a fundamental idea for completing missing exposures, but they require iterative optimization and are prone to producing artifacts. Recently, Kalantari *et al.* [15] introduce an end-to-end neural network consisting of an optical flow prediction model and a fusion network. Chen *et al.* [3] propose a coarse-to-fine alignment framework, which utilizes optical flow for coarse alignment and then employs deformable convolution [7] for fine alignment. Chung *et al.* [6] employ spatial attention instead of optical flow to achieve alignment from adjacent frames to the reference frame. Xu *et al.* [40] propose the HDR-domain loss, which uses optical flow supervised by HDR frames for LDR alignment, yielding favorable results. However, when complex situations such as large motion, saturation, and others occur simultaneously, these methods struggle to achieve precise alignment, resulting in artifacts in the final outputs. Besides, as the exposures of reference frames alternate, the reconstructed HDR frames tend to inherit the defects of reference frames, leading to flicker in the output videos. The fundamental issue lies in the lack of exposure information at every time stamp. Different from

previous methods, we propose a renaissance work which focuses on completing missing exposures for the reference frame in neural HDR video rendering. Through compensating for the defects of the alternatively exposed reference frames by utilizing the completed results, our approach avoids artifacts of ghost and noise and obtains a temporally consistent HDR video result.

## 3 PROPOSED METHODS

### 3.1 Overview

Each frame $l_i$ in the LDR video $L = \{l_i | i = 1, ..., n\}$ is expected to have multiple exposures $\mathbf{E} = \{\epsilon_i | i = 1, ..., z\}$ to cover the whole dynamic range of HDR video $H = \{h_i | i = 1, ...n\}$, where $n$ is the length of video, and $z$ is the number of exposures. However, in our task, only one exposure can be obtained in one certain time stamp, which creates an exposure sequence $E = \{e_i | i = 1, ...n\}$ for the LDR video, where $e_i = \epsilon_{(i \oslash z)+1}$. $a \oslash b$ indicates the remainder of $a$ divided by $b$.

Following previous work [40], we try to render a high-quality HDR video with LDR video under two settings of different alternate exposures: two alternate exposures ($z = 2$) with three frames as input, and three alternate exposures ($z = 3$) with five frames. We use the two-exposure setting as an example to demonstrate how to achieve HDR rendering. In Sec. 3.5, we introduce how to extend our approach to the three-exposure setting. The frame $l_t$ corresponding to the current time stamp $t$, is treated as the reference frame. $l_t$ is combined with its neighbors as input $\{l_{t-1}, l_t, l_{t+1}\}$. To render HDR frames, we explicitly complete frames $\widehat{l_t}$ with absent exposure $\epsilon_i$, where $i \neq (t \oslash z)+1$. Therefore, with complete exposure information, we can render a high-quality HDR video.

Our method consists of two processes: exposure completing for the frame with the absent exposure at the time stamp $t$ and coarse-to-fine HDR rendering. These two processes are coupled together through an encoder-decoder architecture, as illustrated in the Fig. 2. The feature encoder $\mathcal{E}$ takes LDR frames $\{l_{t-1}, l_t, l_{t+1}\}$ as input and generates pyramid features for each frame. As the spatial dimensions decrease and the feature channels increase, the encoder outputs feature sets $\{\Phi_{t-1}, \Phi_t, \Phi_{t+1}\}$ for input frames, each of which is a set of $k$-level pyramid features $\Phi = \{\phi^k | k = 1, 2, 3, 4\}$. Considering the distinct characteristics of the two processes, we devise two dedicated decoders. Initially, an exposure completing decoder $\mathcal{D}_I$ is employed to interpolate frame $\widehat{l_t}$ and its corresponding features $\widehat{\Phi}_t^k = \{\widehat{\phi}_t^k | k = 1, 2, 3\}$. Subsequently, a coarse-to-fine HDR rendering decoder $\mathcal{D}_R$ takes $\{\Phi_{t-1}, \Phi_t, \Phi_{t+1}\}$ and $\widehat{\Phi}_t$ as input, and outputs optical flows $F_{t-1 \rightarrow t} = \{f_{t-1 \rightarrow t}^k | k = 0, 1, 2, 3\}$, $F_{t+1 \rightarrow t} = \{f_{t+1 \rightarrow t}^k | k = 0, 1, 2, 3\}$, coarse HDR features $\widehat{\Psi}_t = \{\widehat{\psi}_t^k | k = 1, 2, 3\}$ and coarse HDR frame $\widehat{h}_t^c$. The optical flows here are utilized to warp $\{\Phi_{t-1}, \Phi_{t+1}\}$ or $\{l_{t-1}, l_{t+1}\}$ to time stamp $t$, yielding warped neighbor features $\widetilde{\Phi}_{t-1} = \{\widetilde{\phi}_{t-1}^k | k = 1, 2, 3\}$, $\widetilde{\Phi}_{t+1} = \{\widetilde{\phi}_{t+1}^k | k = 1, 2, 3\}$ or frames $\{\widetilde{l}_{t-1}, \widetilde{l}_{t+1}\}$. The warped neighbor features $\{\widetilde{\Phi}_{t-1}, \widetilde{\Phi}_{t+1}\}$ and frames $\{\widetilde{l}_{t-1}, \widetilde{l}_{t+1}\}$ are fed into $\mathcal{D}_I$. Subsequently, the exposure completing features $\widehat{\Phi}_t$ from $\mathcal{D}_I$ are fed into the HDR rendering decoder $\mathcal{D}_R$, assisting it in better restoring lost details due to saturation and noise when decoding coarse HDR features $\widehat{\Psi}_t$ and coarse HDR frame $\widehat{h}_t^c$. This design coupling two decoders enables mutual enhancement of the two processes, thus obtaining more accurate exposure completing frame $\widehat{l_t}$ and better coarse HDR result $\widehat{h}_t^c$. To obtain the final HDR rendering result, we map $\{l_{t-1}, l_t, l_{t+1}\}$ and $\widehat{l_t}$ to the linear HDR domain. The function defining the mapping of LDR frames to the linear HDR domain is as follows:

$$x_t = l_t^\gamma / e_t, \tag{1}$$

where $\gamma$ is a hyperparameter and $e_t$ is the exposure time of $l_t$. Finally, the ultimate high-quality HDR rendering result $\widehat{h}_t$ is obtained by fusing the input and completed frames with coarse HDR frame in linear HDR domain using a simple blending network. The details of each process will be presented in the following sections.

## 3.2 Exposure Completing

Under the setting that input LDR frames are with alternating exposures, completing the absent exposure information is essential for rendering HDR frames. However, existing neural HDR rendering methods overlook this essential problem, making it difficult for these methods to accurately recover detailed information when motion occurs with saturation or noise simultaneously, potentially leading to artifacts. Furthermore, due to these methods heavily depend on reference frames, the alternate appearance of noise and saturation caused by the alternate exposures in reference frames affects the temporal consistency of the video. Therefore, to better address the aforementioned issues, we explicitly tackle the fundamental problem of HDR rendering by completing frames with absent exposure corresponding to reference frames.

Given the LDR frame input $\{l_{t-1}, l_t, l_{t+1}\}$ with alternate exposures $\{e_{t-1}, e_t, e_{t+1}\}$, where the reference frame and neighbor frames have two different exposures ($e_t = \epsilon_1$ and $e_{t-1} = e_{t+1} = \epsilon_2$), our task is to complete the frame $\widehat{l_t}$ with absent exposure $\epsilon_2$ corresponding to reference frame through interpolation.

Our exposure completing process takes the neighbor features $\{\Phi_{t-1}, \Phi_{t+1}\}$ and the optical flows $\{F_{t-1 \rightarrow t}, F_{t+1 \rightarrow t}\}$ as input. During this process for the features with absent exposure information, the $k$-th level neighbor features $\{\phi_{t-1}^k, \phi_{t+1}^k\}$ and optical flows $\{f_{t-1 \rightarrow t}^k, f_{t+1 \rightarrow t}^k\}$ are processed to obtain completed feature $\widehat{\phi}_t^k$:

$$\widetilde{\phi}_{t-1}^k, \widetilde{\phi}_{t+1}^k = \mathcal{W}(\phi_{t-1}^k, f_{t-1 \rightarrow t}^k), \mathcal{W}(\phi_{t+1}^k, f_{t+1 \rightarrow t}^k), \tag{2}$$

$$\widehat{\phi}_t^3 = \mathcal{D}_I^3 \left( \left[ \widetilde{\phi}_{t-1}^3, \widetilde{\phi}_{t+1}^3 \right] \right), \tag{3}$$

$$\widehat{\phi}_t^k = \mathcal{D}_I^k \left( \left[ \widetilde{\phi}_{t-1}^k, \widetilde{\phi}_{t+1}^k, \mathcal{U}_2(\widehat{\phi}_t^{k+1}) \right] \right), \tag{4}$$

where $\mathcal{W}(\cdot, \cdot)$ denotes using optical flow to warp neighbor features to the reference feature, $\mathcal{U}_2$ represents the bilinear upsampling operation with scale factor 2, $\mathcal{D}_I^k$ ($k = 1, 2$) represents the middle levels of the exposure completing decoder $\mathcal{D}_I$, and $[\cdot]$ indicates concatenation operation. In the final step of the exposure completing process, the completed frame $\widehat{l_t}$ is explicitly interpolated with neighbor frames $l_{t-1}, l_{t+1}$ and optical flow $f_{t-1 \rightarrow t}^0, f_{t+1 \rightarrow t}^0$ as input:

$$\widetilde{l}_{t-1}, \widetilde{l}_{t+1} = \mathcal{W}(l_{t-1}, f_{t-1 \rightarrow t}^0), \mathcal{W}(l_{t+1}, f_{t+1 \rightarrow t}^0), \tag{5}$$

$$\widehat{l_t} = \mathcal{D}_I^0 \left( \left[ \widetilde{l}_{t-1}, \widetilde{l}_{t+1}, \mathcal{U}_2(\widehat{\phi}_t^1) \right] \right), \tag{6}$$

where $\mathcal{D}_I^0$ represents the highest level of the exposure completing decoder $\mathcal{D}_I$.

As illustrated in the Fig. 1, with the above design, we have obtained realistic exposure completing result for the challenging areas with motion, saturation and noise, providing accurate and complete exposure information for subsequent HDR rendering.

## 3.3 Coarse-to-fine HDR Rendering

Building upon the results of above process, we designed a coarse-to-fine HDR rendering network based on completed features $\widehat{\Phi}_t$ and frame $\widehat{l_t}$. Taking the original features $\{\Phi_{t-1}, \Phi_t, \Phi_{t+1}\}$, completed features $\widehat{\Phi}_t$ and warped neighbor features $\{\widetilde{\Phi}_{t-1}, \widetilde{\Phi}_{t+1}\}$ as input, the HDR rendering decoder $\mathcal{D}_R$ reconstructs HDR features $\widehat{\Psi}_t$ and optical flows $\{F_{t-1 \rightarrow t}, F_{t+1 \rightarrow t}\}$ in a coarse-to-fine manner, and

finally obtains the coarse HDR frame $\widehat{h}_t^c$:

$$\widehat{\psi}_t^3, f_{t-1\to t}^3, f_{t+1\to t}^3 = \mathcal{D}_R^4\left(\left[\phi_{t-1}^4, \phi_t^4, \phi_{t+1}^4\right]\right), \tag{7}$$

$$\widehat{\psi}_t^{k-1}, f_{t-1\to t}^{k-1}, f_{t+1\to t}^{k-1} =$$
$$\mathcal{D}^k\left(\left[\phi_t^k, \widehat{\phi}_t^k, \widehat{\psi}_t^k, f_{t-1\to t}^k, f_{t+1\to t}^k, \widetilde{\phi}_{t-1}^k, \widetilde{\phi}_{t+1}^k\right]\right), \tag{8}$$

$$\widehat{h}_t^c, f_{t-1\to t}^0, f_{t+1\to t}^0 =$$
$$\mathcal{D}^1\left(\left[\phi_t^1, \widehat{\phi}_t^1, \widehat{\psi}_t^1, f_{t-1\to t}^1, f_{t+1\to t}^1, \widetilde{\phi}_{t-1}^1, \widetilde{\phi}_{t+1}^1\right]\right), \tag{9}$$

where $\mathcal{D}_R^k$ ($k = 2, 3$) stand for the $k$-th level of the HDR rendering decoder $\mathcal{D}_R$. The warped neighbor features $\{\widetilde{\phi}_{t-1}^k, \widetilde{\phi}_{t+1}^k\}$ help $\mathcal{D}_R$ to identify regions with poor alignment and subsequently obtain the refined optical flow $\{f_{t-1\to t}^{k-1}, f_{t+1\to t}^{k-1}\}$.

Based on the completed frame $\widehat{l}_t$ and the coarse HDR frame $\widehat{h}_t^c$, we combine them with original LDR frames to obtain the final HDR rendering result $\widehat{h}_t$. We adopted a blending network with U-net architecture from Xu *et al.* [40]. Thus, original and completed LDR frames $\{l_{t-1}, l_t, l_{t+1}, \widehat{l}_t\}$, along with their corresponding frames $\{x_{t-1}, x_t, x_{t+1}, \widehat{x}_t\}$ in linear HDR domain, and coarse HDR frame $\widehat{h}_t^c$ are fed into the blending network. The blending network calculates fusion weights $W = \{w_i | i = 0, 1, 2, 3, 4\}$ for the input five HDR domain frames $\{x_{t-1}, x_t, x_{t+1}, \widehat{x}_t, \widehat{h}_t^c\}$ and obtains the final HDR rendering result $\widehat{h}_t$ through a weighted average based on the computed weights $W$:

$$\widehat{h}_t = \frac{w_0\widehat{h}_t^c + w_1\widehat{x}_t + w_2x_t + w_3x_{t-1} + w_4x_{t+1}}{\sum_{j=0}^4 w_j}. \tag{10}$$

Following Xu *et al.* [40], we also fuse with the neighbor frames $\{x_{t-1}, x_{t+1}\}$ to provide information about static regions. As depicted in Fig. 1, when encountering complex scenes involving both motion and saturation or noise, the fused exposure completion frames effectively provide the missing exposure information, thereby obtaining ghost-free and low-noise HDR results. In summary, based on completing the frame with missing exposure information, we finally achieve a high-quality HDR rendering process.

## 3.4 Training Loss

We calculate the losses for the completed LDR features $\widehat{\Phi}_t$ and frame $\widehat{l}_t$, rendered HDR features $\widehat{\Psi}_t$ and frame $\widehat{h}_t$, and optical flows $\{F_{t-1\to t}, F_{t+1\to t}\}$:

**The losses for images.** The widely adopted $\mathcal{L}_1$ loss is used to supervise the completed LDR frame $\widehat{l}_t$ and rendered HDR frame $\widehat{h}_t$, and is defined as follows:

$$\mathcal{L}_I^{com} = \left\|\widehat{l}_t - \bar{l}_t\right\|_1, \tag{11}$$

$$\mathcal{L}_I^{ren} = \left\|\mathcal{T}(\widehat{h}_t) - \mathcal{T}(\bar{h}_t)\right\|_1, \tag{12}$$

$$\mathcal{L}_I = \mathcal{L}_I^{com} + \mathcal{L}_I^{ren}, \tag{13}$$

where $\bar{l}_t$ and $\bar{h}_t$ are the corresponding ground truth for the completed LDR frame $\widehat{l}_t$ and rendered HDR frame $\widehat{h}_t$, respectively. The $\mathcal{T}(\cdot)$ is a widely used function to map HDR frame to the tone-mapped HDR domain, since HDR images are typically displayed after tone mapping. This simple differentiable $\mu$-law function $\mathcal{T}(\cdot)$ is defined as follows:

$$\mathcal{T}(h) = \frac{\log(1 + \mu h)}{\log(1 + h)}, \tag{14}$$

where $\mu$ is a hyperparameter.

**The losses for features.** Feature space geometry loss proposed in [21] is employed to ensure that the intermediate features obtained from frame interpolation and HDR reconstruction can be more effectively refined to conform to geometrically structured features. The parameter shared encoder $\mathcal{E}$ is used to obtain corresponding pyramid features $\overline{\Phi}_t = \{\overline{\phi}_t^k | k = 1, 2, 3\}$ and $\overline{\Psi}_t = \{\overline{\psi}_t^k | k = 1, 2, 3\}$ from the ground truth of completed LDR and rendered HDR frames. Then, the loss function for supervising the features of completed LDR and rendered HDR frames can be written as:

$$\mathcal{L}_G^{com} = \sum_{k=1}^3 \mathcal{L}_{cen}(\widehat{\phi}_t^k, \overline{\phi}_t^k), \tag{15}$$

$$\mathcal{L}_G^{ren} = \sum_{k=1}^3 \mathcal{L}_{cen}(\widehat{\psi}_t^k, \overline{\psi}_t^k), \tag{16}$$

$$\mathcal{L}_G = \mathcal{L}_G^{com} + \mathcal{L}_G^{ren}, \tag{17}$$

where the $\mathcal{L}_{cen}$ is census loss [32] and computed using the soft Hamming distance between census-transformed [46] feature maps, considering 3×3 patches in a channel-by-channel manner.

**The loss for optical flows.** The HDR-domain alignment loss [40] is used hierarchically to supervise the learning process of optical flow, which is defined as follows:

$$\mathcal{L}_F^{t-1\to t} = \sum_{k=0}^3 \|\mathcal{W}(\mathcal{T}(\bar{h}_{t-1}), \mathcal{U}_{2^k}(f_{t-1\to t}^k)) - \mathcal{T}(\bar{h}_t)\|_1, \tag{18}$$

$$\mathcal{L}_F^{t+1\to t} = \sum_{k=0}^3 \|\mathcal{W}(\mathcal{T}(\bar{h}_{t+1}), \mathcal{U}_{2^k}(f_{t+1\to t}^k)) - \mathcal{T}(\bar{h}_t)\|_1, \tag{19}$$

$$\mathcal{L}_F = (1 - m_t) \odot (\mathcal{L}_F^{t-1\to t} + \mathcal{L}_F^{t+1\to t}), \tag{20}$$

where $\mathcal{W}(\cdot, \cdot)$ denotes using optical flow to warp neighbor frames to the reference frame, $\mathcal{U}_s$ represents the bilinear upsampling operation with scale factor $s$. The mask $m_t$ indicates well-exposed regions in reference frame. First, $l_t$ is converted to YCbCr color space to extract luminance $y$. Then, $m_t$ is defined as $\delta_{\text{low}} < y < \delta_{\text{high}}$, where $\delta_{\text{low}}$ and $\delta_{\text{high}}$ are the low and high luminance thresholds, respectively. This allows the optical flow computation to focus more on regions with poor exposure in the reference frame.

**Total loss.** Our total loss can be summarized as follows:

$$\mathcal{L}_{total} = \mathcal{L}_I + \alpha * \mathcal{L}_G + \beta * \mathcal{L}_F. \tag{21}$$

## 3.5 Extension to Three Exposures

In the alternate exposure setting with three exposures, five LDR frames $\{l_{t-2}, l_{t-1}, l_t, l_{t+1}, l_{t+2}\}$ are used as input to the network for HDR $\widehat{h}_t$ rendering. The five LDR frames are with corresponding exposure sequences $\{e_{t-2}, e_{t-1}, e_t, e_{t+1}, e_{t+2}\}$. In this way, the reference frame has only one certain exposure $e_t = \epsilon_3$. Specifically, the first frame $l_{t-2}$ and the fourth frame $l_{t+1}$ have the same exposure $e_{t-2} = e_{t+1} = \epsilon_1$, while the second frame $l_{t-1}$ and the fifth frame $l_{t+2}$ also share the same exposure $e_{t-1} = e_{t+2} = \epsilon_2$. Therefore, the

Table 1: Quantitative comparisons of our method with other state-of-the-art methods on the Cinematic Video dataset [8].The best and the second best results are highlighted in red and blue, respectively.

| Methods | | 2-Exposure | | | 3-Exposure | | |
|---|---|---|---|---|---|---|---|
| | | PSNR $_T$ | SSIM $_T$ | HDR-VDP-2 | PSNR $_T$ | SSIM $_T$ | HDR-VDP-2 |
| Kalantari13 [17] | 2013-TOG | 37.51 | 0.9016 | 60.16 | 30.36 | 0.8133 | 57.68 |
| Kalantari19 [15] | 2019-CGF | 37.06 | 0.9053 | 70.82 | 33.21 | 0.8402 | 62.44 |
| Yan19 [43] | 2019-CVPR | 31.65 | 0.8757 | 69.05 | 34.22 | 0.8604 | 66.18 |
| Prabhakar [36] | 2021-CVPR | 34.72 | 0.8761 | 68.82 | 34.02 | 0.8633 | 65.00 |
| Chen [3] | 2021-ICCV | 35.65 | 0.8949 | **72.09** | 34.15 | 0.8847 | **66.81** |
| LAN-HDR [6] | 2023-ICCV | 38.22 | 0.9100 | 69.15 | 35.07 | 0.8695 | 65.42 |
| HDRFlow [40] | 2024-CVPR | 39.20 | 0.9154 | 70.98 | 36.55 | 0.9039 | 65.89 |
| HDRFlow [40](+Sintel) | 2024-CVPR | **39.30** | **0.9156** | 71.05 | **36.65** | **0.9055** | 66.02 |
| Ours | | **40.59** | **0.9241** | **73.31** | **37.24** | **0.9102** | **68.36** |

Table 2: Quantitative comparisons of our method with other state-of-the-art methods on the DeepHDRVideo dataset [3]. The result is the weighted average of all results from both dynamic scenes and static scenes in this dataset. The best and the second best results are highlighted in red and blue, respectively.

| Methods | | 2-Exposure | | | 3-Exposure | | |
|---|---|---|---|---|---|---|---|
| | | PSNR $_T$ | SSIM $_T$ | HDR-VDP-2 | PSNR $_T$ | SSIM $_T$ | HDR-VDP-2 |
| Kalantari13 [17] | 2013-TOG | 40.33 | 0.9409 | 66.11 | 38.45 | 0.9489 | 57.31 |
| Kalantari19 [15] | 2019-CGF | 39.91 | 0.9329 | 71.11 | 38.78 | 0.9331 | 65.73 |
| Yan19 [43] | 2019-CVPR | 40.54 | 0.9452 | 69.67 | 40.20 | 0.9531 | 68.23 |
| Prabhakar [36] | 2021-CVPR | 40.21 | 0.9414 | 70.27 | 39.48 | 0.9453 | 65.93 |
| Chen [3] | 2021-ICCV | 42.48 | **0.9620** | 74.80 | 39.44 | **0.9569** | 67.76 |
| LAN-HDR [6] | 2023-ICCV | 41.59 | 0.9472 | 71.34 | **40.48** | 0.9504 | 68.61 |
| HDRFlow [40] | 2024-CVPR | 43.18 | 0.9510 | 77.11 | 40.45 | 0.9530 | 72.30 |
| HDRFlow [40](+Sintel) | 2024-CVPR | **43.25** | 0.9520 | **77.29** | **40.56** | 0.9535 | **72.42** |
| Ours | | **43.44** | **0.9558** | **79.20** | 40.13 | **0.9550** | **76.98** |

exposure completion process takes $\{l_{t-2}, l_t, l_{t+1}\}$ and $\{l_{t-1}, l_t, l_{t+2}\}$ as inputs, generates exposure completing results $\{\widehat{l_t^{\epsilon_1}}, \widehat{l_t^{\epsilon_2}}\}$ through the flow-guided completing process, and obtains coarse HDR results $\widehat{h_t^c}$ through the coarse-to-fine HDR rendering process. A total of fifteen images, including the exposure completing frames and multiple original LDR frames, along with their corresponding frames in linear HDR domain, and the coarse HDR result, are fed into the blending network. This blending network calculates seven weights to fuse the seven HDR domain images in a weighted average manner and obtain the final HDR rendering result. More details are provided in supplementary materials.

## 4 EXPERIMENTS

### 4.1 Experimental Setup

**Datasets.** We utilize synthetic training data generated from the Vimeo-90K dataset [41]. To adapt the Vimeo90K dataset for HDR video reconstruction, we follow prior research [18] to convert the original data into LDR sequences with alternate exposures. To create the ground truth of completed LDR frames, we also generated LDR sequences with missing exposures in the same way. Our framework is tested on the Cinematic Video dataset [8] and DeepHDRVideo

dataset [3]. The Cinematic Video dataset has two synthetic videos from indoor and outdoor scenes. The DeepHDRVideo dataset [3] contains both real-world dynamic scenes and static scenes with random global motion augmentation. The HDRVideo dataset [17] is employed solely for qualitative evaluation, as it lacks ground truth.

**Implementation details.** We implement our approach using PyTorch and conduct experiments on an NVIDIA RTX3090 GPU. We employ AdamW optimizer [20] with $\beta_1 = 0.9$ and $\beta_2 = 0.999$. The learning rate is set to $10^{-4}$. In our experiments, we set $\gamma$ in Eq. 1 to 2.2 and $\mu$ in Eq. 14 to 5000. Following HDRFlow [40], we set $\delta_{\text{low}}$ to 0.2 and $\delta_{\text{high}}$ to 0.8. The weighting hyperparameters $\alpha$ and $\beta$ for the loss function in Eq. 21 are set to 0.01.

**Evaluation metrics.** $\text{PSNR}_T$, $\text{SSIM}_T$ and HDR-VDP-2 [30] are adopted as the evaluation metrics. $\text{PSNR}_T$ and $\text{SSIM}_T$ are computed on the tone-mapped images. HDR-VDP-2 is computed with the number of pixels per visual degree set to 30, which means the angular resolution of the image.

### 4.2 Comparisons with State-of-the-art

**Quantitative comparisons** between our and other state-of-the-art methods on the Cinematic Video [8] and DeepHDRVideo [3] datasets are shown in Table 1 and Table 2, respectively. Compared to

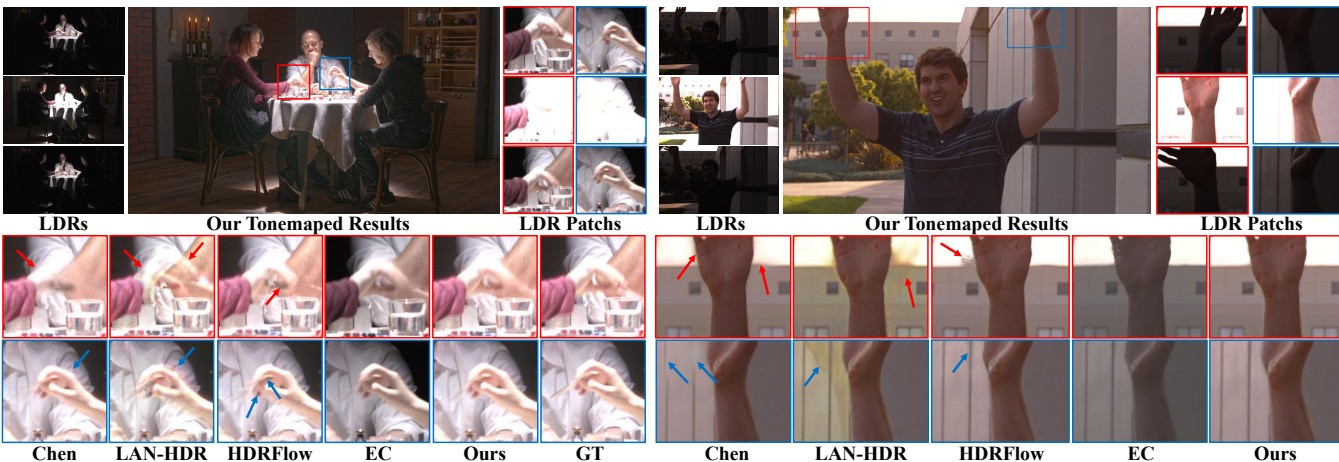

**Figure 3: Qualitative comparison in scenes with over-saturation and motion. Left: 2-Exposure scene from the Cinematic Video dataset [8]. Right: 2-Exposure scene from the HDRVideo dataset [17]."EC" refers to our exposure completion results.**

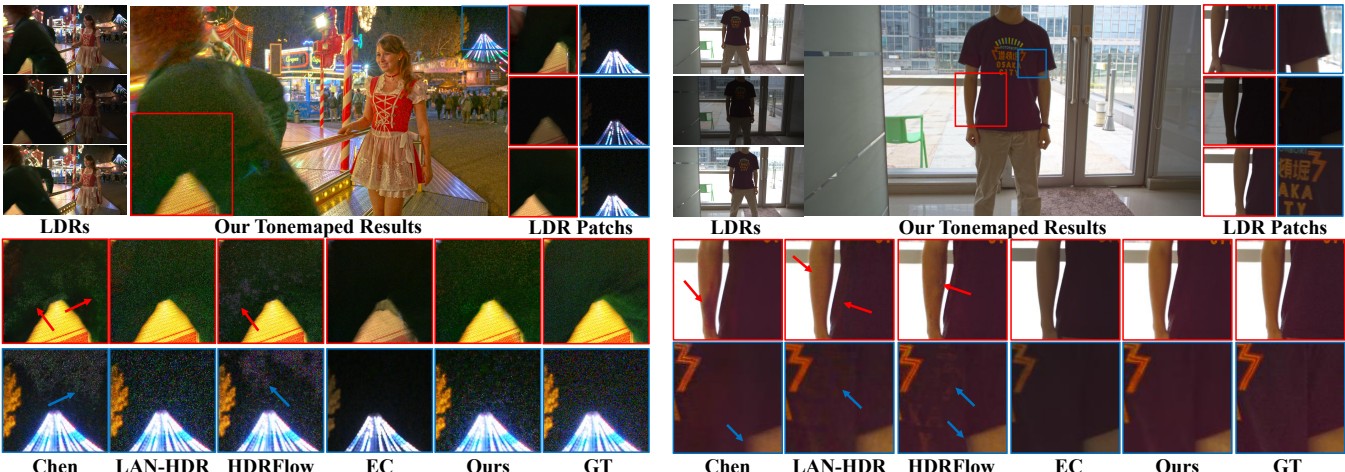

**Figure 4: Qualitative comparison in scenes with noise and motion. Left: 2-Exposure scene from the Cinematic Video dataset [8]. Right: 2-Exposure scene from the DeepHDRVideo dataset [3]. "EC" refers to our exposure completion results.**

state-of-the-art methods, our approach consistently achieves superior or comparable performance. Especially, our approach achieves state-of-the-art performance on Cinematic Video [8] dataset, outperforming the second-best method by 1.29dB and 0.59dB in terms of $PSNR_T$ for the 2-exposure and 3-exposure settings, respectively. **Qualitative comparisons** are shown in Fig. 3 and Fig. 4. we compare our NECHDR with the previous methods: Chen [3], LAN-HDR [6] and HDRFlow [40]. Fig. 3 illustrates the results from scenes with saturation and motion on Cinematic Video dataset [8] and HDRVideo dataset [17] under the 2-exposure setting. Apart from our method, other methods tend to exhibit severe artifacts or detail loss when saturation and motion occur simultaneously. In such challenging scenarios, we achieve accurate and artifact-free exposure completion results for saturated regions by leveraging frame interpolation from neighboring frames. This enables us to fuse high-quality HDR results. And in Fig. 4, we also show the

results encountering noise and motion from the Cinematic Video dataset [8] and DeepHDRVideo dataset [3]. The scene on the left side of Fig. 4 is captured in low-light conditions, resulting in very low signal-to-noise ratio in the low-exposure frames. This can lead to very high noise levels in rendered HDR frames after tone mapping, which makes this scene particularly challenging. Specifically, methods based on optical flow [3, 40] tend to produce noisy artifacts, while attention-based method [6] exhibit more pronounced noise. Our method explicitly reconstructs the absent high-exposure frame with low noise at the current time stamp, achieving the best noise suppression, even with some regions having noise intensity lower than ground truth. The visualization results above explain why we achieve a significant performance improvement compared with other state-of-the-art methods. More qualitative comparisons in the 3-exposure setting are provided in the supplementary materials.

**Table 3: Ablation study of NECHDR on the DeepH-DRVideo [3] and CinematicVideo dataset [8]. "EC" refers to exposure completing. "HR" refers to HDR rendering. "Uncoupled" means this two processes work independently.**

| Model | DeepHDRVideo | | Cinematic Video | |
|---|---|---|---|---|
| | $PSNR_T$ | $SSIM_T$ | $PSNR_T$ | $SSIM_T$ |
| EC Baseline | 41.09 | 0.9479 | 39.13 | 0.9174 |
| HR Baseline | 42.64 | 0.9524 | 39.98 | 0.9193 |
| Uncoupled EC, HR | 43.17 | 0.9535 | 40.27 | 0.9202 |
| Coupled EC, HR (Ours) | **43.44** | **0.9558** | **40.59** | **0.9241** |

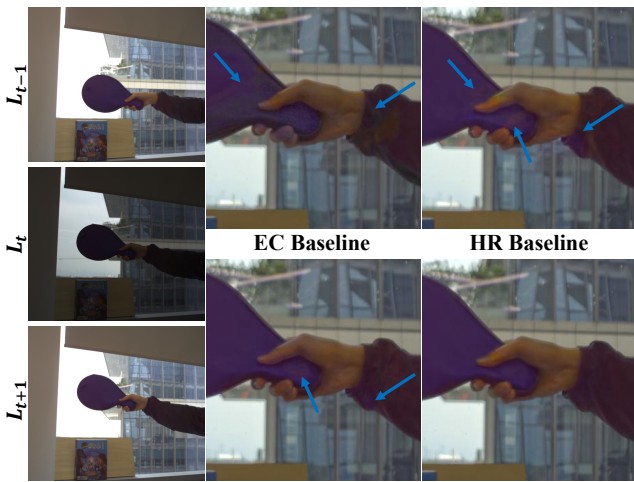

Figure 5: Qualitative comparison of the models corresponding to the ablation study on a dynamic scene of DeepH-DRVideo [3] dataset.

## 4.3 Analysis

**Ablation Study.** We conduct ablation experiments under the 2-exposure setting on Cinematic Video dataset [8] and DeepHDRVideo dataset [3], and the quantitative and qualitative results are shown in Table 3 and Fig. 5, respectively. We devised a baseline that employs IFRNet [21] to utilize the neighbor frames to complete the middle time frame that with missing exposure information and achieve HDR results by simply fusing the completed LDR frame with the original LDR frames. The performance of this baseline is shown as "EC Baseline" in the first row in Table 3. Another baseline directly renders HDR results based on IFRNet, which is the "HR Baseline" in second row in Table 3. However, relying solely on either exposure completing or HDR rendering results is with limited performance. By adding the exposure completing decoder, we get a network (third row in Table 3) that contains both decoupled exposure completing and HDR rendering processes. Finally, through feeding the completed features and frames from exposure completing decoder into the process of HDR rendering, we achieve our NECHDR framework in the forth row in Table 3. Based on the qualitative and quantitative results, we can see: (a) coupled exposure completing and HDR rendering processes benefit the quality

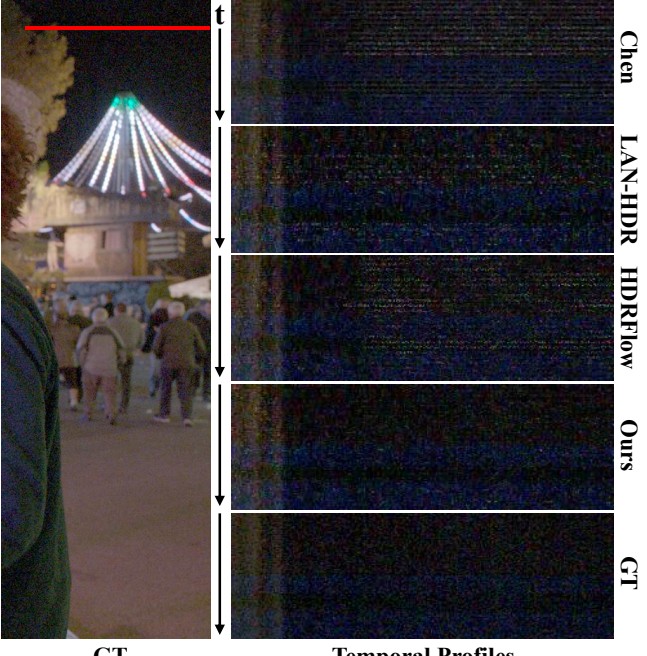

Figure 6: Visual comparisons of temporal consistency.

of final HDR results; (b) exposure completing decoder is necessary; (d) the completed results with missing exposure information help the rendering process of HDR reconstruction.

**Temporal consistency.** We show the visual comparisons of temporal consistency in Fig. 6. In Fig. 6, we record a two-pixel-height line traversing all frames of a scene in the Cinematic Video dataset [8] over time and lay them out sequentially to form temporal profiles. Base on the illustration of temporal profiles, we can observe that the horizontal stripes exist in the temporal profiles of other methods. The horizontal stripes comes from the differences between adjacent frames, which represents the temporal inconsistency. In contrast, the horizontal stripes can hardly be observed in our temporal profiles, which means that our proposed method achieves better temporal consistency. Additional visual comparisons regarding temporal consistency can be found in the supplementary materials.

## 5 CONCLUSION

In this paper, we implement the idea of exposure completing for neural HDR video rendering and propose the Neural Exposure Completing HDR (NECHDR) framework. The NECHDR leverages interpolation of neighbor LDR frames to complete missing exposures, providing a complete set of exposure information for each time stamp. This process of exposure completing creates a novel neural HDR video rendering pipeline, which can generate results of less noise and ghosting artifacts, thereby enhancing temporal consistency. Experimental results on multiple public benchmarks demonstrate the superiority of our NECHDR, which may shift the focus of researchers in this area to the exposure completing.

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
