# OpenReview forum: "Exposure Completing for Temporally Consistent Neural High Dynamic Range Video Rendering"
_acmmm.org/ACMMM/2024/Conference — MM2024 Poster_

### Official Review · Reviewer_q5Xf · 2024-05-21

**Rating:** 4
**Confidence:** 4

**Summary:**

This paper proposes an HDR video rendering approach named Neural Exposure Completing HDR (NECHDR) that explicitly predicts a missing exposure image before fusion to improve reconstructed HDR image.

**Strengths:**

The key modules include an exposure completing decoder that incorporates neighboring LDR frames to predict absent exposures at each time frame guided by predicted optical flows. This is coupled with an HDR rendering decoder that fuses the completed exposures with the original LDRs to produce the HDR output via a coarse-to-fine process. Experiments show NECHDR outperforming prior work in terms of HDR-VDP-2 metrics while reducing ghosting artifacts.

**Limitations:**

This work introduces a two-stage process for reconstructing HDR content from bracketed LDR images. Optical flow is utilized in reconstructing the coarse HDR image before it undergoes processing by the Blending Network. However, the blending network has to address misaligned LDR images and linear HDR images to mitigate ghosting artifacts in the final output, which may be inefficient.

The proposed training pipeline incorporates several loss functions; however, ablation studies analyzing the effectiveness of these loss functions are absent.

Crucial information such as the number of parameters, inference speed, or FLOPS (or equivalent metrics) of the entire network is necessary for evaluating the effectiveness of the proposed method. This information is currently missing for comparison with existing methods.

PSNR, SSIM and HDR-VDP-2 are image quality metrics.  A more comprehensive evaluation would benefit from assessing video quality metrics, such as FovVideoVDP or ColorVideoVDP.

**Suitability:**

2

---

### Official Review · Reviewer_B1Mg · 2024-05-23

**Rating:** 3
**Confidence:** 3

**Summary:**

This paper presents an HDR video reconstruction method based on alternating LDR exposures as input.

**Strengths:**

1. The task is of practical importance.

2. The quantitative results look good.

**Limitations:**

1. The authors emphasize temporal consistency as a key feature of the proposed method. However, the reviewer does not find any specific designs for promoting it in network architecture or loss function.

2. What are the key differences between the architectural designs and other DL-based HDR video reconstruction methods? It seems like the proposed method follows a two-stage approach, which has been extensively explored in the field.

3. The overall loss as the combination of many losses looks ad hoc.

4. The authors are encouraged to use more recent and perceptually relevant loss functions such as HDR-VDP-3, PU-encoded PSNR, and SSIM with and without CRF correction.

5. A psychophysical experiment is suggested to quantify the perceptual gains of the proposed method.

6. For video-based applications, the authors are encouraged t

**Suitability:**

3

---

### Official Review · Reviewer_xjRn · 2024-05-23

**Rating:** 4
**Confidence:** 4

**Summary:**

1. The paper proposed a method for HDR video reconstruction based on the exposure Completing.
 the complete set of exposure information is useful to reduce noise and ghosting artifacts and then provides solid information for the HDR reconstruction. The paper shows the effectiveness of the proposed method, outperforming other promising HDR video models. It shows the ghosting artifacts and the noise can be effectively eliminated with the proposed method.

**Strengths:**

1. The paper is well organized and easily followed.

2. It evaluates their proposed method in various datasets and show the effectiveness of the proposed method, which are convincing.

**Limitations:**

1. The processing time, model complexity, and GFLOPs in the evaluation results should be provided. And also the comparisons with the previous methods also should be provided.

2. There is no loss for the coarse HDR frame \hat(h_t^c) when training. If there is a loss for the coarse HDR frame, would the performance be better?

3. The value for the losses for features. It is easy to understand the image loss and the optical flow loss. But what is the value for the features loss? Ablution study could be performed to verify the effectiveness of this loss.

4. In introduction section line 139-142, the authors claim that the previous alignment based methods rely on the reference image while the exposure of reference frame changes at every time stamp, which means that the reference frames of different exposures may have different defects. But the exposure completing and the coarse HDR image generation are also relied on the reference image. So this part could be refined.

**Suitability:**

2

---

### Official Review · Reviewer_LhpW · 2024-05-24

**Rating:** 2
**Confidence:** 3

**Summary:**

While the idea of completing missing exposure frames to improve temporal consistency in neural HDR video rendering is new and interesting, there are several issues that need to be addressed.

**Strengths:**

1. The proposed idea of completing missing exposure frames is well-motivated and inspired by traditional optical HDR imaging systems.
2. The experiments demonstrate the effectiveness of the proposed method in reducing ghosting and noise artifacts, leading to improved temporal consistency.

**Limitations:**

1. The details of the technical approach are occasionally challenging to comprehend, particularly due to dense notations and equations that lack sufficient explanations.
2. It is recommended to evaluate the influence of varying exposure patterns (for instance, configurations involving more than three exposures) on the performance of the proposed method.
3. A comprehensive ablation study is necessary to delineate the contributions of different framework components, particularly focusing on the role of various loss functions.
4. Figures 3 and 4 reveal that the proposed method introduces noticeable halo artifacts around high-contrast regions, such as around the arms in Figure 4 (right).
5. Although the paper emphasizes temporal consistency, it lacks quantitative comparative analysis in the temporal domain, which is critical for substantiating the claims made.
6. Include a runtime analysis and discussion on the computational complexity of the proposed method.

**Suitability:**

1

---

### Meta-Review · Area_Chair_ikr9 · 2024-06-27

**Recommendation:** Accept (Poster)
**Confidence:** 3

**Metareview:**

The authors argued that existing HDR video rendering method generate flickering results. To address this issue, they proposed to render HDR frames via completing the absent exposure information. It is believed this operation can benefit “the fusing process for HDR results, reducing noise and ghosting artifacts therefore improving temporal consistency”.  Generally, this method proposed is straightforward and the results (including the video presented in supplementary) seem good. This paper is also well organized and easy to read.

After rebuttal, the authors have well addressed most of the reviewers’ concerns. A remaining issue is its insufficient validation of temporal consistency. Since this paper claims temporal consistency as its major contribution while the other methods “generate flickering HDR results”, it is necessary to provide enough experiments to validate this statement. However, the paper fails to present any experiment. In Table 2 of the rebuttal file, the authors provide a rough comparison with three methods: Chen, LAN-HDR and HDRFlow, evaluated by ForVideoVDP.  However, it is still far less than all 8 methods compared in this work. Even in this table, the authors just proved that the proposed method generates better quality than its peers, but it does not support the above claim: the proposed method can guarantee temporal consistency while the others cannot. It is strongly recommended that the authors address this important consideration in their future work.